# LONG-TERM 3D POINT TRACKING BY COST VOLUME FUSION

## ABSTRACT

Long-term point tracking is essential to understanding non-rigid motion in the physical world better. Deep learning approaches have recently been incorporated into long-term point tracking, but most prior work predominantly functions in 2D. Although these methods benefit from the well-established backbones and matching frameworks, the motions they produce do not always make sense in the 3D physical world. In this paper, we propose the first deep learning framework for long-term point tracking in 3D that generalizes to new points and videos without requiring test-time fine-tuning. Our model is a coarse-to-fine approach that contains a cost volume fusion module at each level, which effectively integrates multiple past appearances and motion information via a transformer architecture, significantly enhancing overall tracking performance. Our explicit occlusion reasoning also allowed tracking through long occlusions. In terms of 3D tracking performance, our model significantly outperforms simple scene flow chaining and previous 2D point tracking methods, even if one uses ground truth depth and camera pose to backproject 2D point tracks in a synthetic scenario.

## 1 INTRODUCTION

Motion estimation is a task that has existed since the beginning of computer vision. Short-term dense motion estimation problems, such as optical flow in 2D and scene flow in 3D, have been extensively studied. However, the utility of these tasks is limited because many points are featureless, making their motion estimation within two frames fundamentally ambiguous. Besides, chaining 2-frame motions to derive point trajectories is susceptible to significant cumulative errors and ineffective for handling long occlusions. However, spatio-temporal tracking of keypoints (Laptev, 2005) have always been interesting because it offers the capability to track long-term non-rigid object motions. Such knowledge would be greatly helpful for augmented reality and robotics applications, as well as providing supervision for generative models that generate dynamic videos with arbitrary non-rigid motions. Recently, Harley et al. (2022) reinvigorated the long-term pixel tracking problem and proposed a framework inspired by previous state-of-the-art optical flow and object tracking work. Doersch et al. (2022) released datasets specifically designed to address point tracking. These datasets, including 2D and 3D data, have boosted research on this task.

Most existing methods predominantly address the problem of 2D long-term point tracking. But in the 3D world we live in, such 2D tracking, however accurate it might be, might still miss the 3D motion of the points. Even the state-of-the-art video generator SORA has significant trouble understanding long-term 3D point motion (Bupe, 2024), which may limit its usage for downstream tasks such as augmented reality and robot manipulation. As we will show in the experiments, backprojecting 2D long-term point tracks into 3D, even with known camera poses and pixel depths, is still prone to errors especially at object boundaries, where a difference of 1-2 pixels may lead to a large difference in 3D.

The ability to track any point long-term in 3D would significantly enhance our understanding of the scene dynamics. Recently, Luiten et al. (2023) proposed a test-time optimization method to model dynamic scenes using a set of Gaussians, thereby enabling long-term point tracking. While surpassing the performance of prior 2D methods, this approach relies on test-time optimization for each scene and struggles to track new points entering the scene. Test-time optimization is quite computationally expensive, especially for longer videos, making it unsuitable for online tracking.

In this paper, we propose an efficient and **generalizable** method for long-term **online** tracking of keypoints in a dynamic 3D point cloud. To our knowledge our approach is the first 3D point-based approach that directly attacks long-term 3D point tracking in a generalizable manner that can work on new points and videos **without** test-time fine-tuning. Our online tracking framework takes as input a sequence of point clouds representing the dynamic scene. The model predicts a position for each query point by combining multiple past appearance information with motion information from the past point trajectory with a transformer-based framework. Occlusions are predicted explicitly to filter out noisy appearance features. We propose an adaptive decoding module that selectively decodes around the query point, enabling the network to process denser point clouds and generate more precise motion for each point. Experiment results show that our approach significantly outperforms baselines such as linking scene flow results or backprojection from 2D point tracks.

In summary, our contributions include:

- We propose the first online 3D point-based tracking framework that can track any point in 3D point clouds **without** test-time optimization.
- We devise a novel Cost Volume Fusion module that effectively takes into account the long-term appearances of each point and its past motion trajectory.
- We propose an adaptive decoding module that significantly reduces memory consumption when training on denser point clouds.

## 2 RELATED WORK

### 2.1 POINT TRACKING

Tracking any pixel or point in 2D long-term has recently gained significant attention. MFT (Schmidt et al., 2023) presents an extension for optical flow by constructing optical flows not only between consecutive frames but also between distant frames. These flows are chained together guided by the predicted occlusion and uncertainty scores obtained from pre-trained networks, to derive the most reliable sequence of flows for each tracked pixel. PIPs (Harley et al., 2022) presents a novel framework designed for multi-frame point tracking which simultaneously estimates the target point positions in multiple frames. Therefore, it can handle short occlusion events. To improve tracking performance, Cotracker (Karaev et al., 2023) utilizes self-attention layers to track target points and their local and global contextual points together. The self-attention layers enable information exchange among these points, leading to better tracking quality. Recently, SpatialTracker (Xiao et al., 2024) and SceneTracker (Wang et al., 2024) utilizes a 2.5D approach which collects 3D neighborhoods of 2D pixels using a triplane approach. However, their final output is still within each camera frame and hence predicts a mixture of camera motion and pixel motion.

Tap-Vid (Doersch et al., 2022) offers valuable datasets along with a straightforward baseline method. It achieves this by predicting the position of a point in each frame based on the cost volume derived from the query feature and the corresponding feature map. Meanwhile, TAPIR (Doersch et al., 2023) introduces a two-stage network comprising a matching stage and a refinement stage. This framework also incorporates the prediction of uncertainty to suppress ambiguous or unreliable predictions, enhancing point tracking accuracy. Bozic et al. (2020) introduces a differentiable non-rigid approach that achieves superior performance in reconstructing non-rigidly moving objects. However, this approach can only focus on a single object in the video. In contrast, we focus on modeling entire dynamic scenes by tracking any point within them.

A different line of work utilizes test-time optimization techniques to model the scene, resulting in superior tracking performance. Additionally, these methods directly track points in 3D. Therefore, they can utilize useful 3D priors for tracking purposes. Specifically, OmniMotion (Wang et al., 2023) represents video content by employing a canonical 3D volume. It learns a set of bijections to map points between any frame and the canonical one, thereby enabling tracking capability. Note that the 3D canonical volume here is only used for finding 2D point correspondences between frames and does not represent the real-world coordinate system, limiting OmniMotion to 2D point tracking. Similarly, Luiten et al. (2023) represents the scene with a set of Gaussians. While the number of Gaussians, their colors, and opacity remain fixed throughout the video, these Gaussians can move and rotate freely to model the dynamic scene. Therefore, the tracking capability emerges from persistently modeling the dynamic scene under these constraints.

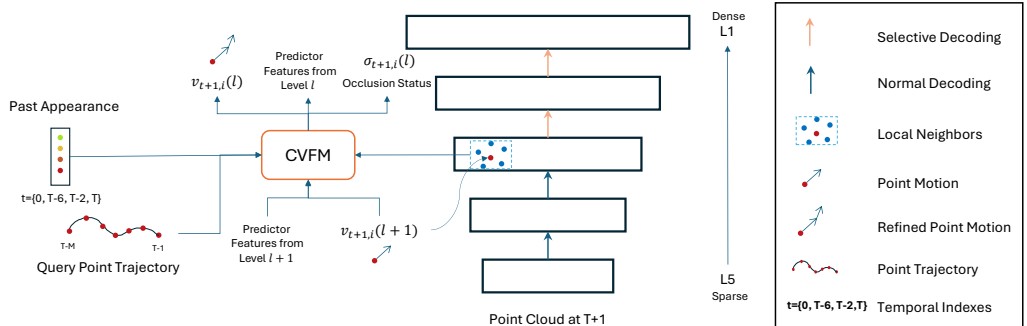

Figure 1: **Long-term Tracking Framework**. Given a sequence of point clouds as input, we use a U-Net based backbone to extract the point cloud's features hierarchically. For simplicity, only the decoder branch is shown. At each level, we refine the sparse motion from the previous level of the backbone for the query point by jointly considering multiple past appearances and the past motion of the query. The motion predicted at level 1 is used as the final motion of the query point from frame t to t+1.

To achieve superior tracking performance, Wang et al. (2023); Luiten et al. (2023) require significant time to reconstruct the 3D model of the scene using test-time optimization. Consequently, these methods cannot be used for online tracking. In contrast, our method tracks points online without test-time optimization.

## 2.2 SCENE FLOW ESTIMATION

Scene flow estimation predicts the 3D motion field of points, with the input being either 2D images or 3D point clouds. We focus on the approaches that has 3D point clouds as input as they are more relevant to our work. Approaches can usually be categorized into two types: supervised and self-supervised. However, many supervised methods can utilize self-supervised losses to adapt a pre-trained model to a new dataset without ground truth.

FlowNet3D (Liu et al., 2019) is among the first to use PointNet++ (Qi et al., 2017), a deep network for point clouds, to predict scene flow from point clouds directly and proposes important basic modules such as flow embedding, set conv, and set upconv layers that are commonly used in subsequent works. FLOT (Puy et al., 2020) reformalizes scene flow estimation as an optimal transport problem to initially compute scene flow and subsequently refine it using a deep network. PointPWC (Wu et al., 2020) introduces novel cost volumes, upsampling, and warping layers and utilizes a point convolution network (Wu et al., 2019) to handle 3D point cloud data. Scene flow is then constructed in a coarse-to-fine fashion. The authors also propose a novel self-supervised loss that has been adopted in subsequent works (Zhang et al., 2024a; Wang et al., 2021; Fang et al., 2024; Shen et al., 2023). PV-Raft (Wei et al., 2021) introduces a point-voxel correlation field to handle both long-range and local interactions between point pairs. NSFP (Li et al., 2021b) represents scene flow implicitly with a neural network trained directly on test scenes with self-supervised losses. Fast neural scene flow (Li et al., 2023) improves upon NSFP by utilizing the distance transform (Breu et al., 1995; Danielsson, 1980) and a correspondence-free loss, significantly reducing processing time. Finally, with the rise of diffusion models, Zhang et al. (2024b); Liu et al. (2024) incorporate diffusion processes into the scene flow estimation pipeline, achieving millimeter-level end-point error.

## 3 METHOD

### 3.1 OVERVIEW

We aim to track any point in a video of a dynamic 3D scene. We assume camera poses and depth information have been obtained (e.g. from a SLAM system (Mur-Artal & Tardós, 2017)). With these, the video is converted to a sequence of point clouds $V = \{p_0, \ldots, p_T\}$ where $p_t = \{p_{t,i}\}$, and $p_{t,i} \in R^3$ denotes the 3D coordinates of each point $i$ in the point cloud at time step $t$. Let $q_{t,j} \in R^3$ be the $j^{th}$ query point at $t$. For each query point $q_{t,j}$, the model predicts the 3D motion

$v_{t+1,j} = q_{t+1,j} - q_{t,j}$ and the occlusion status $\sigma_{t+1,j} \in \{0,1\}$ in the next frame. The same process is then repeated autoregressively for long-term point tracking.

The multi-level features for each scene point are obtained using a point-based U-Net backbone (Wu et al., 2020), comprising an encoder and a decoder. Let $p_t(l)$, $f_t^{p,E}(l)$, and $f_t^{p,D}(l)$ represent the point cloud used at level $l$ and its corresponding encoder/decoder features extracted at level $l$ from our backbone where $l = 1$ represents the densest level and $l = L$ the sparsest level. Here, $p_t(l)$ is derived by applying grid-subsampling on $p_t(l-1)$ with $p_t(1) = p_t$. For simplicity, unless explicitly stated, we illustrate the algorithm on a single level and hence refer to the point cloud and their features as $p_t$ and $f_t^p$. Unless specified, the features $f_t^p$ represent the decoder features $f_t^{p,D}$. We use $f_{t,i}^p$ to denote the decoder feature at a specific point location $p_{t,i}$.

## 3.2 BACKGROUND - POINTPWC

PointPWC (Wu et al., 2020) is a deep network that can be trained in either a supervised or unsupervised fashion to predict the scene flow between two point clouds. In PointPWC, a patch-to-patch strategy is used to obtain a robust cost volume and increase the receptive field.

$$C(p_{t,i}) = \sum_{p_{t,u} \in N_t(p_{t,i})} W_t(p_{t,i}, p_{t,u}) \sum_{p_{t+1,j} \in N_{t+1}(p_{t,u})} W_{t+1}(p_{t,u}, p_{t+1,j}) cost(p_{t,u}, p_{t+1,j}) \quad (1)$$

$$W_t(p_{t,i}, p_{t,u}) = MLP(p_{t,i} - p_{t,u})$$

where $N_t(p)$ represents a neighborhood around $p$ at time $t$, and $cost(p_{t,u}, p_{t+1,j})$ refers to the matching cost between two points. The cost is computed via MLP using the concatenation of color features at $p_{t,u}$ and $p_{t+1,j}$ and the relative position between the points as input (Wu et al., 2020):

$$cost(p_{t,u}, p_{t+1,j}) = MLP([f_{t,u}^p, f_{t+1,j}^p, p_{t+1,j} - p_{t,u}]).$$

The idea of the cost volume is to incorporate matching between points in the neighborhood $N_t(p_{t,i})$ and the neighbors of each point in the frame $t+1$, which is similar to a small 2D window that is commonly used to derive 2D cost volumes (Sun et al., 2018) – it increases the range that points $p_{t,i}$ can match to as well as the robustness of the matching.

Given the cost volume $C(p_{t,i})$ of a point $p_{t,i}$ at level $l$, the PointPWC framework estimates the flow for each point in a coarse-to-fine manner. Specifically, given the predictor features from level $l+1$, the framework first upsamples them to points on level $l$, then concatenates them with the cost volume $C(p_{t,i})$, as well as the decoder feature $f_{t,i}^p$ at level $l$. These are used together by the flow predictor to generate the predictor features and the residual scene flow at level $l$. The latter is then summed with the upsampled scene flow from level $l+1$ as the predicted flow at level $l$.

## 3.3 COST VOLUME FOR LONG-TERM POINT APPEARANCE

One difference between our long-term point tracking framework and scene flow is the existence of query points that are not necessarily within the given point cloud. Given the hierarchical decoder features of the point cloud, $f_t^p$, the feature of the query point $q_{t,i}$ can be extracted by applying a PointConv layer as follows:

$$f_{t,i}^q = MLP\left(\sum_{p_{t,j} \in N(q_{t,i})} W(q_{t,i}, p_{t,j}) f_{t,j}^p\right). \quad (2)$$

Since a single point does not contain enough appearance information, when talking about the appearance of each query point $q_i$, it should be understood as the appearance of the local region containing that point, which could deform from time to time. By jointly considering multiple appearances from different time, tracking performance can be improved. Specifically, given a set of appearances of a query point $q_i$ up to the time step $t$, $F_{t,i}^q = \{f_{t_1,i}^q, f_{t_2,i}^q, \ldots, f_{t_n,i}^q\}$, where $t_1, t_2, \ldots, t_n$ represent the frames storing the query's appearances ($t_n \leq t$. Refer to Sec. 3.4), we can obtain a set of cost volumes $C_{t,i}^q = \{C_{t_1}(q_{t,i}), \ldots, C_{t_n}(q_{t,i})\}$ as follows:

$$C_{t_k}(q_{t,i}) = \sum_{p_{t+1,j} \in N(q_{t,i})} W(q_{t,i}, p_{t+1,j}) cost_{t_k}(q_{t,i}, p_{t+1,j}) \quad (3)$$

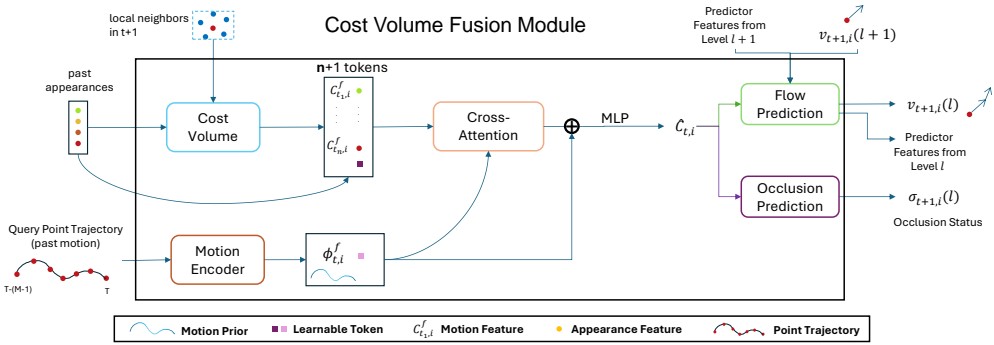

Figure 2: **Cost Volume Fusion Module**. We propose a novel Cost Volume Fusion Module to predict the query point motion by jointly considering multiple appearances and the past motion trajectory of the query. These appearances are used to compute a set of cost volumes, which are combined with the motion prior via cross-attention in the transformer layer, followed by an MLP. The output features from the MLP are subsequently used to predict the refined motion and the occlusion status of the query point.

$$cost_{t_k}(q_{t,i}, p_{t+1,j}) = MLP([f^q_{t_k,i}, f^p_{t+1,j}, p_{t+1,j} - q_{t,i}])$$

where we use the feature at timestep $t_k$ to account for the query's multiple appearances.

Note that here we used a simpler cost volume construction without the patch-to-patch formulation as in Eq. (1). This is because for long-term tracking, it is difficult to define neighbors at frame $t + 1$ from a past frame $t_k$ that could be very far apart temporally. Results in Table 1 show that our cost volume formulation is only slightly less effective than the patch-to-patch formulation.

Another important aspect of the query point features is that based on Eq. (2), they are only affected by scene points surrounding the query point. Therefore, we propose to **selectively decode** only the points surrounding the query points and prune other points to reduce total computation and memory consumption, especially during training time. We only use this selective decoding strategy for $l \in \{1, 2\}$, which are the two densest levels and thus have the highest decoder memory requirements. By utilizing selective decoding, we increased the number of points per frame that fit into GPU memory from $8,192$ to $60,000$ (with 16 frames used for each mini-batch during training), significantly improving the performance of the algorithm, particularly its capability of making predictions with sub-pixel accuracy. For scene flow, this recovered the lost ground we had with the simpler cost volume formulation (Table 1), and ablations for long-term point tracking can be found in the supplementary materials.

### 3.4 FUSION OF APPEARANCE AND MOTION CUES

A major benefit of long-term point tracking over simply chaining 2-frame scene flows is the capability to incorporate motion cues. Motion cues can help produce motion estimates during occlusion or a blurry frame. However, due to our goal of non-rigid point tracking, points can move in surprising and different ways that cannot be easily captured by a motion prior. In those cases, a good appearance tracking module should take over and predict more precise locations.

To properly consider multi-frame appearance and motion jointly, we devise a novel data-driven Cost Volume Fusion module that softly combines motion-based and appearance-matching-based features. The output of this combination is then used to predict the actual motion and occlusion status for each point in the target frame. Below, we detail the specific components of the module.

#### 3.4.1 COST VOLUME FUSION MODULE

To provide the motion prior for the network, the last $M$ predicted motions of the query point, $v_{(M,t),i} = [v_{t-(M-1),i}, \ldots, v_{t,i}]$, are concatenated and encoded with an MLP followed by a group normalization layer (Wu & He, 2018) to obtain a motion prior vector $\phi_{t,i} = MLP(v_{(M,t),i})$. Note that, at the beginning of the video, the list of past motions $v_{(M,t),i}$ is initialized with zeros. Addi-

tionally, each level $l$ utilizes a separate MLP to encode the motion prior while using the same list of past motions as input.

For each appearance feature of the query extracted at $t_k$, we calculate the corresponding cost volume $C_{t_k,i}$ (a simplified notation for $C_{t_k}(q_{t,i})$) using Eq. (3). Intuitively, the cost volume contains the appearance matching information the query point in the current timestep and its corresponding neighboring points in the next timestep - Fig. 2. Therefore, the cost volume provides appearance matching information to predict the potential location of the query point in the next timestep. More details about cost volume can be found at Sec. 3.2 and Eq. 3. In the experiment, to estimate the motion of a query point from $T$ to $T+1$, we extract the appearance of the query point from the following frames, $\{0, T-6, T-2, T\}$ to reduce the appearance redundancy from consecutive frames.

The cost volume module is shown in Fig. 2. To predict the motion of the $i^{th}$ query in the current frame at the sparse level $l$, our network jointly uses the motion and appearance information extracted from multiple time steps in the past. Specifically, unlike approaches for short-term point correspondence or scene flow, we maintain a list of potential appearance vectors for each query point as mentioned above and calculate its corresponding cost volume with Eq. (3).

Besides, we also consider pure motion-based information from our motion prior $\phi_{t,i}$. This information is especially beneficial when the query point is not visible, as well as other cases where the appearance is ambiguous.

Each cost volume $C_{t_k,i}$ is concatenated with the corresponding point appearance $f^q_{t_k,i}$ to obtain $C^f_{t_k,i}$. It contains information on the matching between points from the frame $t+1$ and past query point appearances. The motion prior $\phi_{t,i}$ is concatenated with a learnable feature token. We propose to use the motion prior as the query and to cross-attend to all $\{C^f_{t_k,i}\}^n_{k=1}$. The intuition is to select the points that match some past query point appearance while are also plausible w.r.t. the motion trajectory. Besides, we also use a learnable token $E$ to allow the module to rely on the motion prior when no appearance-matching information is available due to occlusion. This module can be implemented by stacking multiple transformer decoder (Vaswani et al., 2017) layers together.

$$O_{t,i} = \text{Cross-Attn}(\{C^f_{t_1,i}, \dots, C^f_{t_n,i}, E\}, \phi_{t,i})$$
$$\hat{C}_{t,i} = MLP(O_{t,i} + \phi_{t,i})$$

Finally, to predict the motion at the current level $l$, $v_{t,i}$, we use the flow predictor head from Wu et al. (2020) which takes the transformer features $\hat{C}_{t,i}$ and the predictor features from level $l+1$ as input.

Besides, the transformer features $\hat{C}_{t,i}$ are fed to an MLP to predict the occlusion status at each level $l$. By training this, we encourage the model to store the occlusion information within the cost volumes. However, during inference, we only use the predicted occlusion at level 1 as the final occlusion prediction for each query point.

### 3.5 MODEL TRAINING

Our model relies on the estimated frame-to-frame motion of each point to construct the long-term point trajectory. Therefore, instead of directly training the model on the long-term tracking data, we split the training process into two stages:

- **Scene flow pretraining**. The whole model is pre-trained with the scene flow datasets. Each training sample includes two consecutive frames randomly sampled from a video. After the training, the model can achieve competitive performance in the scene flow estimation task.
- **Long-term tracking**. In this stage, the Cost Volume Fusion Module is added to handle multiple appearances of the query point and its past trajectory. The network is trained with randomly sampled longer videos.

By following this two-stage training pipeline, we can utilize synthetic scene flow datasets to improve the overall tracking performance and stabilize the training process. This has been important for the model to achieve good performance.

We supervise the model by the GT point position and the GT scene flow as follows:

$$L^{track} = \frac{1}{Tn_q} \sum_{l=1}^{L} \sum_{t=1}^{T} \sum_{i=1}^{n_q} \alpha^{l-1} |q_{t,i} - \hat{q}_{t,i}|_1$$

$$L^{sf} = \frac{1}{T} \sum_{l=1}^{L} \sum_{t=1}^{T} \sum_{i=1}^{|p_t|} \gamma^{l-1} |\Delta p_{t,i} - \Delta \hat{p}_{t,i}|_1$$

where $q_{t,i}$ and $\hat{q}_{t,i}$ are the predicted and the ground truth positions, and $\Delta p_{t,i}$ and $\Delta \hat{p}_{t,i}$ are the predicted and the ground truth flow of the scene point $p_{t,i}$.

Motion in the 3D world is usually smooth in terms of both direction and magnitude unless the target is affected by external force from a collision. To encourage such smoothness property, we introduce a smoothness loss that minimizes the difference between the predicted motions in consecutive frames of each query point. The motion smoothness can be defined over all query points as follow:

$$L^{smooth} = \frac{1}{LTn_q} \sum_{l=0}^{L} \sum_{i=1}^{n_q} \sum_{t=0}^{T-1} ||v_{t,i} - v_{t+1,i}||_1 \qquad (4)$$

We also attempted to use $L^{rigid}$ and $L^{iso}$, the rigidity and isometry losses from Luiten et al. (2023), although our ablations will show that they do not lead to a significant difference in performance. Altogether, we train the model using the weighted sum of the above losses:

$$L = \lambda_1 \cdot L^{sf} + \lambda_2 \cdot L^{track} + \lambda_3 \cdot L^{smooth} + \lambda_4 \cdot L^{rigid} + \lambda_5 \cdot L^{iso}$$

We use grid-search to find the optimal values for these hyper parameters. During the scene flow pretraining stage, we only use $L^{sf}$ and $L^{track}$. $\lambda_1$ and $\lambda_2$ are set to 2 and 1 respectively. During the second stage, we set $\lambda_1 = 2$, $\lambda_2 = 1$, $\lambda_3 = 0.3$, and $\lambda_4 = \lambda_5 = 0.2$ for all datasets.

## 4 EXPERIMENTS

### 4.1 DATASET AND TRAINING DETAILS

We use the FlyingThings (Mayer et al., 2016) dataset to pre-train the scene flow model, and then train and test on two separate datasets, TAPVid-Kubric (Doersch et al., 2022) and PointOdyssey (Zheng et al., 2023).TAPVid-Kubric is a synthetic dataset with 9,760 training videos and 250 testing videos. Each video has 24 frames with resolution $256 \times 256$. In each validation video, 256 query points are randomly sampled from all frames. The model is required to track these points in the rest of the video. In the training set, we can generate an arbitrary number of query points for training. Because the dataset is synthetic, we can generate ground truth depth for all points.

We first build a scene flow task based on the point tracking ground truth on TapVid-Kubric's training split and fine-tune the pretrained model on this task. Then, with the encoder and decoder backbone frozen to save GPU memory, the full model is fine-tuned on 16-frame videos from the point tracking dataset constructed from TapVid-Kubric (Doersch et al., 2022). For data augmentation, we use random horizontal flipping, random scaling of the point-cloud coordinates, and random temporal flipping. We use a batch size of 16 during the scene flow pre-training stages and a batch size of 8 during the training of the full model.

**PointOdyssey** provides much longer synthetic videos (over 1,000 frames and up to 4,000 frames) for training and testing. The dataset includes 131/15/13 videos in the training/validation/testing split. Because Point Odyssey does not provide scene flow ground truth, we augment a single frame with random translations and rotations to simulate scene flow data. The model is first fine-tuned on these simulated scene flow data before being trained on the entire training set. We observe that the model supervised with the simulated scene flow data tends to converge faster in the second phase than the one trained with self-supervised loss.

We utilized a U-Net-based PointConvFormer (Wu et al., 2023) backbone. This is similar to PCF-PWC in Table 1 but with our simplied cost volume (Eq. (3)) instead of theirs (Eq. (1)). During the inference stage, we have query points that can appear in any place in the video. Hence, we run the model twice (forward and backward) for each video to track the query points in both directions.

| Methods | EPE3D(m)↓ |
|---|---|
| PointPWC (Wu et al., 2020) | 0.0588 |
| PCFPWC (Wu et al., 2023) | **0.0416** |
| PV-RAFT (Wei et al., 2021) | 0.0461 |
| FLOT (Puy et al., 2020) | 0.0520 |
| HCRF-Flow (Li et al., 2021a) | 0.0488 |
| HPLFlowNet (Gu et al., 2019) | 0.0804 |
| FlowNet3D (Liu et al., 2019) | 0.1136 |
| **Ours - 8k points** | 0.0509 |
| **Ours - 60k points** | **0.0399** |

Table 1: Scene flow estimation on the FlyingThing dataset

## 4.2 METRICS

We extend previous 2D point tracking metrics to 3D:

- Occlusion accuracy (OA) is the accuracy of the occlusion prediction for each query point on each frame.
- $\delta^x$ measures the position accuracy of the predicted point on each frame where the point is visible. A predicted point is considered correct if it is within $x$ centimeters (*cm*) from the ground truth position.
- $\delta^{avg}$ is the average of $\delta^x$ with $x \in [1, 2, 4, 8, 16]$ (*cm*).

For PointOdyssey where the videos are longer, we also adopt the survival rate metric (Luiten et al., 2023) (SR) which is the average number of frames before each tracked point drifts $T$ *cm* away from the ground truth position, divided by the number of frames in the video, with $T = 50$.

We also report results with 2D metrics in order to compare with other 2D methods, but those are secondary results because the primary goal of this paper is 3D tracking.

## 4.3 SCENE FLOW PRE-TRAINING RESULTS

In Table 1, we show the scene flow pretraining results on the FlyingThings dataset. Our framework outperforms PointPWCNet and other 2-frame baselines. Our scene flow performance was significantly improved when we used an input point cloud size of $60,000$ points over the conventional $8,000$ points used in Wu et al. (2020) despite our simpler cost volume computation than PointPWC-Net and PCF-PWCNet.

## 4.4 3D & 2D EVALUATIONS

To our knowledge no prior deep learning-based work tackled the problem of generalizable 3D point tracking. Hence, we mainly compare with 2D baselines and simple scene flow chaining. Since the dataset is synthetic, we can lift 2D tracking results to 3D using the ground truth camera pose and depth map. Due to the subpixel-level predictions from the 2D point trackers, the depth of each point is obtained by interpolation on the provided depth map. For the occluded points, the depth is linearly interpolated using the depth of that point before and after the occlusion. Results with the alternative nearest neighbor interpolation are similar and are shown in the supplementary material.

Table 2: Results on 3D Kubric. Best results are shown in red and second best in blue. (Best viewed with color)

| | OA | 3D - $\delta^{avg}$ | 3D - $\delta^{avg}_{occluded}$ |
|---|---|---|---|
| TAPIR | **96.5** | 46.9 (**-26.9**) | 3.8 |
| CoTracker | 92.5 | **57.8 (-16.0)** | **8.7** |
| Scene Flow Chaining | - | 45.6 (**-28.2**) | 5.6 |
| **Ours** | **93.4** | **73.8** | **44.4** |

Table 2 shows 3D point tracking results comparing our proposed approach with state-of-the-art 2D approaches TAPIR (Doersch et al., 2023) and CoTracker (Karaev et al., 2023), as well as simply chaining the pretrained scene flow. We significantly outperform both approaches by 15.3% and 26.2% points on the $3D - \delta^{avg}$ metric. The performance difference is more significant in the

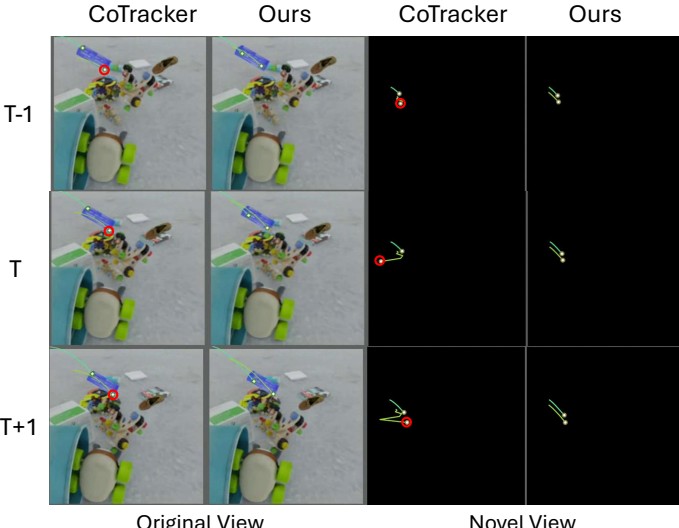

Figure 3: **Qualitative Results**. We reproject the results of CoTracker into 3D and back-project that into a different view point. One can see that because of small errors in 2D leading the CoTracker result on the red circled point off the blue object at time $T$, it incurs significant 3D errors which can be seen as a sudden jump in the trajectory if rendered from a novel viewpoint. (Best viewed in color)

Table 3: Results on 3D Point Odyssey.

|  | 128 Frames | | 512 Frames | | Full Seq | |
|---|---|---|---|---|---|---|
|  | SR | $\delta^{avg}$ | SR | $\delta^{avg}$ | SR | $\delta^{avg}$ |
| PIPS++ (Zheng et al., 2023) | 64.4 | 35.0 | 39.7 | 26.4 | 16.0 | 14.0 |
| **Ours** | **91.0** | **65.1** | **82.6** | **52.0** | **68.5** | **35.0** |

occluded areas, where we record a $44.0\%$ accuracy whereas TAPIR and CoTracker obtain accuracies lower than $10\%$ due to not having a good 3D motion prior to maintain a good track during occlusion. Note that baseline results were already generated by interpolating tracks using the **ground truth** depth. Better nonlinear interpolation may improve their performance by a bit, but it is unlikely that their performance would catch up to our approach.

Such significant performance differences support our arguments that even accurate 2D tracking can have significant issues locating accurate tracks in 3D, even with fully known camera pose and ground truth depth. A qualitative illustration in Fig. 3 indicates the issue with 2D trackers. At frame $T$, although the original view does not indicate a significant error on the red circled point that is being tracked, the tracking actually drifted slightly off the blue object. In consequence, if we render it from a novel viewpoint outside of the original 2D image plane, we can see a significant jump in the trajectory. At time $T + 1$, the tracked point went back to the blue object and from the novel view we again see a significant jump in the trajectory. This indicates significant errors in 3D tracking despite

Table 4: Results on 2D Kubric. *: reproduced results. Best results are shown in red and second best in blue. TAPIR and CoTracker used an hourglass network that re-runs the encoder and decoder several times, whereas our method does not use hourglass and is comparable with their results with a single iteration

|  | OA | 2D - $\delta^{avg}$ | 2D - AJ | 2D - $\delta^{avg}_{occluded}$ |
|---|---|---|---|---|
| COTR (Jiang et al., 2021) | 78.55 | 60.7 | 40.1 | - |
| RAFT (Teed & Deng, 2020) | 86.4 | 58.2 | 41.2 |  |
| PIPs (Harley et al., 2022) | 88.6 | 74.8 | 59.1 | - |
| Tap-Net (Doersch et al., 2022) | 93.0 | 77.7 | 65.4 | - |
| TAPIR* (1 iter) (Doersch et al., 2023) | 94.6 | 88.8 | 81.0 | 27.5 |
| CoTracker* (1 iter) (Karaev et al., 2023) | - | 90.0 | - | 56.9 |
| Scene Flow Chaining | - | 77.0 | - | 25.8 |
| **Ours** | **93.4** | 87.8 | 75.4 | **61.2** |
| TAPIR* (Doersch et al., 2023) | **96.5** | **92.9** | **86.1** | 36.8 |
| CoTracker* (Karaev et al., 2023) | 92.5 | **93.9** | **84.5** | **66.0** |

Table 5: Ablation on Appearance and Motion Priors

| | OA | 2D - $\delta^{avg}$ | 2D - $\delta^{avg}_{occluded}$ | 3D - $\delta^{avg}$ | 3D - $\delta^{avg}_{occluded}$ |
|---|---|---|---|---|---|
| Multi-App + Motion Prior | **93.4** | **87.0** | **60.8** | **73.1** | **44.0** |
| Multi-Appearance | 92.9 | 86.0 | 57.9 | 71.6 | 40.5 |
| Single-Appearance | 91.3 | 83.5 | 52.7 | 61.6 | 29.8 |

Table 6: Ablation on Regularizations

| | OA | 2d - $\delta^{avg}$ | 2d - $\delta^{avg}_{occluded}$ | 3d - $\delta^{avg}$ | 3d - $\delta^{avg}_{occluded}$ |
|---|---|---|---|---|---|
| Ours | 93.4 | 87.0 | 60.8 | 73.1 | 44.0 |
| Rigid & Isometry | 93.4 | 86.8 | 60.9 | 73.1 | 44.4 |
| Smoothness | 92.4 | 83.0 | 55.3 | 55.3 | 30.0 |

low 2D tracking errors. Our approach, on the other hand, works naturally in 3D. Hence it does not suffer from such drifting issues and produces much more consistent 3D tracking results.

Results on the test split of PointOdyssey in Table 3 show a similar trend. We measure the Survival Rate and $\delta^{avg}$ on the first 128 frames and 512 frames of each video or the full sequences. Our framework also outperforms the baselines by a large margin across all the metrics.

In Table 4, we show results on the 2D evaluation on Kubric. We projected our 3D results to 2D using the known camera parameters for our approach. Our approach outperformed many baselines and are generally comparable or slightly worse than TAPIR and CoTracker in 2D when only a single iteration is used for them. Note that for our 3D tracking results in Table 2, we only compared against the full version of TAPIR and CoTracker (i.e., with multiple iterations) and still outperformed them. We did not include the OA and 2D-AJ (Average Jaccard) numbers for CoTracker (1 iter) because CoTracker produces the occlusion status only at the last iteration. TAPIR and CoTracker additionally utilize an hourglass network that decodes, re-encodes and decodes several times to obtain more precise predictions, but that is against the spirit of online algorithms and we did not pursue that path. We did outperform TAPIR significantly in the prediction of occluded points, showing the benefits of interpolation of the motion from 3D.

## 4.5 Ablation Experiments

We analyze the contribution of using multiple appearance features and the motion prior. Results are presented in Table 5. We demonstrate that by employing multiple appearances and the motion prior, the performance across all 2D and 3D metrics consistently improves over using a single appearance. Additionally, the performance improvement under occlusion surpasses that in normal conditions, showing that utilizing multiple appearances of the query point makes the model more resilient to occlusion. By integrating appearances from the past, the model can recover from a few bad appearances during or right before/after occlusion and achieve better results.

We also conduct ablation on the regularization terms. Results in Table 6 show that our smoothness term is very useful in terms of improving the 3D point tracking results. The rigidity and isometry loss from Luiten et al. (2023) provide marginal improvements in the 2D results.

## 5 Conclusion

In this paper, we proposed a 3D long-term point tracking approach based on fusing multiple cost volumes and motion information with a transformer model, which, to the best of our knowledge, is the first generalizable online long-term 3D point tracking approach using deep learning. By selective decoding, we significantly increased the size of the input point cloud that fits into GPU memory, which improves the performance of scene flow and 3D long-term point tracking. In terms of 3D point tracking performance, our approach significantly outperforms scene flow chaining and 2D long-term point tracking approaches even if they are backprojected to 3D with ground truth depths and camera poses, showing the benefits of tracking in 3D. We hope this paper could increase the interest of the community in 3D long-term point tracking. In future work, we plan to utilize our 3D point tracking framework in downstream tasks.

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

## A  APPENDIX

You may include other additional sections here.

