# OpenReview forum: "Long-Term 3D Point Tracking By Cost Volume Fusion"
_ICLR.cc/2025/Conference — Submitted to ICLR 2025_

### Official Review · Reviewer_SZMC · 2024-11-02

**Soundness:** 3
**Presentation:** 2
**Contribution:** 3
**Rating:** 5
**Confidence:** 4

**Summary:**

The paper presents a novel deep-learning method for long-term point tracking in 3D, which generalizes to new points and videos without test-time fine-tuning. Using a coarse-to-fine approach with cost volume fusion modules and transformers, the model effectively integrates appearance and motion information, allowing it to handle occlusions and outperform prior 2D methods and scene flow chaining in 3D tracking accuracy.

**Strengths:**

1. The paper tackles an important problem in computer vision.

2. The proposed method outperforms the 2D tracking and Scene Flow Chaining baselines in 3D point tracking, and surpasses scene flow methods in scene flow estimation on the FlyingThings dataset.

3. The authors provide ablation studies of their design choices.

**Weaknesses:**

1. The paper could benefit from improved writing, especially in the sections describing the method.

2. The method claims to be the first online deep learning-based tracking framework capable of tracking any point in 3D point clouds. However, both SpatialTracker [a] (CVPR24) and SceneTracker [b] also propose generalizable 3D tracking methods. The authors should cite, discuss, and compare their method to these works.

3. While you propose a learning-based method, comparing it with optimization-based 3D trackers as a reference would be beneficial.

Note: You should define the EPE3D metric, which is used in Table 1, in the text.

[a] Xiao, Yuxi, et al. "SpatialTracker: Tracking Any 2D Pixels in 3D Space." Proceedings of the IEEE/CVF Conference on Computer Vision and Pattern Recognition. 2024.

[b] Wang, Bo, et al. "SceneTracker: Long-term Scene Flow Estimation Network." arXiv preprint arXiv:2403.19924 (2024).

**Questions:**

Please see the weaknesses.

---

> ### Author Response · Authors · 2024-11-28
>
> **Weakness #1: The paper could benefit from improved writing, especially in the sections describing the method.**
>
> Please leave us comments on the parts that are unclear in the final comments. We will update the final version to reflect the comments.
>
>
> **Weakness #2: The authors should cite, discuss, and compare their method to SpatialTracker and Scene Tracker**
> Please see the common response for a comparison against SpatialTracker. SceneTracker is an unpublished concurrent work with ours, and it is similar to SpatialTracker as both are iterative, non-online  2.5D approaches that estimate in the camera coordinate frame rather than the world coordinate frame. We’ll cite it as well.
>
> **Weakness #3: comparing the learning-based approach with optimization-based 3D trackers for a reference**
>
> Please see the common response for why such a comparison is not possible.

---

### Official Review · Reviewer_iJKS · 2024-11-03

**Soundness:** 2
**Presentation:** 3
**Contribution:** 2
**Rating:** 6
**Confidence:** 4

**Summary:**

This paper presents a feed-forward approach for tracking points in 3D space across long video sequences. While previous methods mostly work in 2D and can produce physically implausible motions when projected to 3D, this method directly tracks in 3D space and works without test-time optimization. The key technical contributions are a coarse-to-fine approach with cost volume fusion at each level (using transformers to combine past appearance and motion information) and explicit occlusion handling. The authors show their method outperforms both scene flow chaining and 2D tracking methods that are projected into 3D, even when those methods have access to ground truth depth and camera poses (all experiments performed on synthetic data).

**Strengths:**

- I like the paper's idea and the problem it tackles. The problem setting is quite close to scene flow - while there have been many scene flow papers recently, it remains an unsolved problem. The paper's main novelty lies in its network architecture. Although its individual components aren't novel (the overall architecture resembles recent point tracking papers like PIPs, Harley et al.), applying it to long-term 3D tracking is a nice contribution.

- The paper is well-written and well-structured, with clearly described technical details. I especially appreciate the well-written related work section, which helps readers who don't work directly in scene flow/tracking.

- The paper demonstrates good quantitative improvements over SOTA methods.

**Weaknesses:**

* Missing evaluations on real data: Although the paper has extensive evaluations against recent scene flow methods, it lacks any evaluation or even qualitative results on real data (I checked the supplementary material as well). While I understand that real data doesn't always provide good ground truth, especially for dynamic objects, making evaluation challenging, the absence of results on common benchmarks like the KITTI Scene Flow dataset is unfortunate. The complete lack of real-data results, even qualitative ones, makes it difficult to assess the paper's practical impact. I would like to see more results on real data in the rebuttal.

* Missing discussion about online tracking: The paper doesn't compare to test-time optimization methods like Wang et al. (2023) and Luiten et al. (2023), arguing that their method "tracks points online without test-time optimization." While I agree it's a feed-forward method that's potentially less computationally expensive than test-time optimization, "online" is a strong claim. There's no discussion or computational cost analysis comparing their method to existing approaches. (There is some runtime information in the supplementary, but no comparison against other methods).

**Questions:**

I'm optimistic about the paper overall, but would like to see more results on real data to complement the synthetic data presented in the paper.

Final review after rebuttal: I thank the authors for addressing some of my concerns and correcting some terminology misunderstanding. That being said, I agree with other reviewers about the writing quality could be improved, and I'm not completely convinced about the novelty of the proposed method. Given these reasons, I improved the rating to 6 (slightly above acceptance).

---

> ### Author Response · Authors · 2024-11-28
>
> We thank the reviewer for the insightful feedback. Your questions about real-world results and comparisons with test-time optimization methods are answered in the common questions.
>
> Below are answers to your questions in addition to the common answers:
>
>
> **Weakness #2: Online is a strong claim.**
>
> The reviewer may have some misunderstanding about the term “online”. “Online” as used in common computer vision literature merely means that the method makes predictions depending only on **current and past information**, which is true for our algorithm but not true for many baselines, e.g. SpatialTracker,  OmniMotion and Dynamic Gaussian Splatting which uses the entire sequence to make predictions of earlier frames (offline, as they use information from the **future** for every frame).

---

### Official Review · Reviewer_HSRg · 2024-11-04

**Soundness:** 3
**Presentation:** 3
**Contribution:** 2
**Rating:** 5
**Confidence:** 3

**Summary:**

The paper presents a novel deep learning framework for long-term 3D point tracking, leveraging cost volume fusion with a transformer architecture. Designed to track dynamic points without test-time optimization, the model incorporates multiple past appearance and motion cues, outperforming existing 2D tracking approaches when projected to 3D.

**Strengths:**

Utilizes a transformer-based cost volume fusion module to handle occlusions and integrate long-term appearance and motion information.

Extensive performance evaluation demonstrates superiority over 2D methods, especially in occluded scenarios.

**Weaknesses:**

As the paper claimed that it is the first method to achieve 3D point tracking without test-time optimization, to the reviewer's best of knowledge, some works such as FlowNet3D could also predict point cloud tracking results without test-time optimization. The difference between the proposed method and these methods is not clear.

As the paper strengthens that it does not need test-time optimization, a brief comparison of runtime performance would enhance the practical applicability discussion.

The model relies heavily on synthetic datasets (e.g., TAPVid-Kubric, PointOdyssey), which may limit generalizability to real-world, varied environments.

The paper conducts qualitative comparisons only with CoTracker, which is insufficient and somehow weak. It may need to include more qualitative comparisons in the paper.

**Questions:**

There are some unclear expressions in the paper. For example, what is the meaning of (-26.2) (-15.3) in Table 2? It is not clear why novel view in Figure 3 is black and inconsistent with the original view color.

---

> ### Author Response · Authors · 2024-11-28
>
> We thank the reviewer for the insightful feedback. Your questions about real-world results and qualitative results are answered in the common questions, hence we would hope the reviewer would check those. In terms of the TapVID-Davis dataset, that dataset does not have any 3D ground truth and we didn’t claim to improve the 2D performance over the 2D long-term tracking baselines, hence we decided to test on the ADT dataset instead.
>
>
> Below are answers to your questions in addition to the common answers:
>
>
> **Weakness #1: Difference with FlowNet3D is not clear**
>
> FlowNet3D is a scene flow prediction framework that estimates the short-term motion of each point between 2 frames. The goal of this paper is long-term point tracking, which attempts to generate long-term point trajectories that are more consistent than simply chaining scene flow predictions. We developed our own scene flow model, which significantly outperforms FlowNet3D for the 2-frame scene flow task (Table 1). The long-term tracking results by chaining 2-frame scene flow models are presented in Table 2 under the label Scene Flow Chaining, which is significantly worse than our model.
>
> **Weakness #2: brief comparison of runtime performance**
>
> We benchmark following methods on the ADT dataset where each video has 300 frames.
> | Method    | FPS|
> | -------- | ------- |
> | Ours | 3.6 |
> | Spatial Tracker | 3.7 |
>
> For the OmniMotion (Wang et al. 2023) method, each video could take **several hours** even on A100 due to the optimization during test time, which is several orders of magnitude slower than our method. Besides, OmniMotion is actually not a true 3D method as it can only output 2D tracks, please see the common answer.
>
> **Question #1: what is the meaning of (-26.2) (-15.3) in Table 2?**
>
> We are sorry for the confusion. They mean that the results of these respective methods are 26.2 and 15.3 percentage points lower than the results of our method. Those numbers show the difference from our previous best number (73.1). When we achieved the current best number (73.8), we only updated the table with 73.8. We have updated the difference as well in the table to reflect our current result.
>
> **Question #2: It is not clear why novel view in Figure 3 is black and inconsistent with the original view color.**
>
> We did not render the novel view scenes and only visualized the point trajectories with a black background. This is because in the Kubric dataset, the depth map from a single view at a specific timestep allows for only partial scene reconstruction. Consequently, the rendered image of the partial scene in a new view may appear noisy. Therefore, we didn’t render the novel views and only visualized the point trajectories in that view.

---

### Official Review · Reviewer_45rn · 2024-11-04

**Soundness:** 3
**Presentation:** 2
**Contribution:** 2
**Rating:** 5
**Confidence:** 4

**Summary:**

This paper introduces a new deep learning framework specifically designed for long-term 3D point tracking, capable of functioning without test-time optimization. Recognizing the limitations of previous 2D tracking methods, the authors have developed a coarse-to-fine tracking approach that leverages a transformer architecture and a cost volume fusion module at each level of processing. This allows for effective integration of multiple past appearances and motion information, significantly enhancing tracking accuracy and performance, particularly through periods of occlusion.

**Strengths:**

1. The paper introduces a deep learning-based framework for long-term 3D point tracking, a first in its domain according to the authors' claims. This approach is impressive how it combines various techniques to address a complex problem.
2. The authors provided extensive experimental results over various benchmarks to verify their claims.

**Weaknesses:**

1. The authors claim this is the first method for 3D point tracking. However, a highly related work SpatialTracker [A] is neither discussed or compared here. Although SpatialTracker works on 2D images and depths instead of directly on 3D points, its input and output are exactly the same to this work. The authors should discuss and compare with SpatialTracker to verify their points. This is also related to another question, "is it really necessary to lift depths to 3D points for 3D tracking?"

2. The reviewer is quite curious about the 2D projection accuracy of the method. The paper provides Table 4, but this is not very convincing because Kubric is a super simple synthetic dataset and the proposed model has been trained on Kubric. The reviewer recommends testing the proposed method on TAP-Vid DAVIS or other real-world datasets with complex point tracks and occultation to verify its effectiveness.


[A] SpatialTracker: Tracking Any 2D Pixels in 3D Space. Xiao, Yuxi and Wang, Qianqian and Zhang, Shangzhan and Xue, Nan and Peng, Sida and Shen, Yujun and Zhou, Xiaowei. CVPR 2024.

**Questions:**

Reviewers would recommend that the authors provide necessary comparison to SpatialTracker and test the proposed method on TAP Vid DAVIS.


(Not important): the paper mentions "then train and test on two separate datasets, TAPVid-Kubric (Doersch et al., 2022) and PointOdyssey (Zheng et al., 2023).". Does this mean the model was trained on the combination of Kubric and PointOdyssey?

---

> ### Author Response · Authors · 2024-11-28
>
> We thank the reviewer for the insightful feedback. Most of the questions the reviewer asked were  responded in the common feedback. Hence we would hope the reviewer would check those. In terms of the TapVID-Davis dataset, that dataset does not have any 3D ground truth and we didn’t claim to improve the 2D performance over the 2D long-term tracking baselines, hence we decided to test on the ADT dataset instead.
>
> Below are answers to your questions in addition to the common answers:
>
> **Question #2: Was the model trained on the combination of Kubric and PointOdyssey?**
>
> The model was trained on Kubric training set for the Kubric experiment. It is then fine-tuned on the PointOdyssey dataset before tested on the corresponding testing split for the PointOdyssey experiment.

---

### Author Response · Authors · 2024-11-28

We thank the reviewers for their valuable feedback. Below, we address common questions and have marked updates in the paper and supplementary document in red. Please see our individual responses for other queries.

**Q1. Comparison with Spatial Tracker, and whether it’s necessary to lift depths to 3D points for 3D tracking**

Thanks for pointing out the reference, which we have now cited. SpatialTracker operates in 2.5D, outputting query point locations in the next 2D frame, combining camera and point motion. In contrast, our method works in 3D world coordinates, outputting motion in world space. Additionally, SpatialTracker uses iterative estimation and future information, making it offline, while our approach relies only on past data.

We would argue SpatialTracker is still primarily a method based on 2D matching, with the additional triplane encoder providing some 3D neighborhood information. Such methods have difficulties keeping still points to be still in featureless regions as their output mixes camera motion and point motion. Although one can tune ARAP (As-Rigid-As-Possible) parameters to improve on that, it is difficult to balance the need of less rigidity constraints for points with deformable motion and the need for more rigidity constraint for still points. Hence, there would be a tendency to either oversmooth deformable motion, or to have the still points always jitter around. The separation between moving and still points is easier to deal with once we have an explicit 3D approach, where still points should have a motion of zero. See also our results below for this effect in a real-world dataset. Indeed, one constraint of the true 3D approach such as ours is the need to know camera pose. But nowadays there are many SfM and SLAM approaches that obtain excellent camera pose estimates and one can use the outputs from those systems.

**Q2. Real world results**

We present results on the real-world Tapvid3D-ADT dataset (minival split), detailed in Section 8 of the supplementary. The dataset includes ground truth depths from 3D scanning and estimates from ZoeDepth, a monocular depth estimator with an unknown, frame-dependent scaling factor, complicating frame-to-frame reconciliation. SpatialTracker did not test 3D tracking on real datasets for this reason (as noted in their paper). For a fair comparison, we use ground truth depth data for both methods.


| Method    | 3D-AJ $\uparrow$| APD $\uparrow$| OA $\uparrow$
| -------- | ------- | ------- | ------- |
| Ours (fine-tuned on one part of ADT) | 49.9   | 63.5 | 88.5 |
| Ours (trained only on Tapvid)  | 11.7 | 25.0 | 56.1 |
| Spatial Tracker | 10.1 | 14.9 | 72.6

The results show that even our method trained only on the synthetic Kubric dataset outperforms SpatialTracker, whereas our results when fine-tuned on a small part of ADT improves the performance further significantly. By tracking points explicitly in the world coordinate system rather than the camera coordinate system, we effectively disentangle camera motion from actual point motion, enabling a clear distinction between dynamic and static points. Qualitatively, we see that with SpatialTracker the static points cannot stay static and keep jittering around, which is a main reason that led to their lower scores.

**Q3: Comparison to test-time optimization approaches such as Wang et al (2023) and Luiten et al (2023)**

We did not use Dynamics 3D Gaussian Splatting (Luiten et al. 2023) as our baseline because it cannot be applied to the 3D point tracking benchmarks such as Kubric, PointOdyssey, and TAPVid-3D, where only a **monocular** video is provided as input for the point tracking task. Dynamics 3DGS only uses depth information to initialize the set of Gaussians. During the optimization process, it introduces **many new** Gaussians and relies on **multiple views** with a sufficiently large baseline apart from each other to reason about 3D properties of these Gaussians. In addition to that, Dynamic 3DGS assumes that all scene points are visible in the first frame with their multi-camera setup. As a result, after initializing 3D Gaussians in the first frame, they only optimize for Gaussian motions in the subsequent frames without having any capability of handling new points not appearing in the first frame, making this work not applicable to the 3D point tracking benchmarks mentioned above.

Unlike Dynamics 3DGS, OmniMotion (Wang et al 2023) does not output tracks in 3D. It utilizes a 3D space called a quasi-3D canonical volume for accurate 2D tracking, but this 3D space cannot be used to convert their 2D tracks into 3D tracks in the real world coordinate system, making them ineligible for the 3D point tracking evaluation. We updated the main paper to clarify this difference.

**Q4. Qualitative Results**

We added 2 more videos (*kubric_comparison.mp4* and *odyssey_comparison.mp4*). More discussion about these new qualitative results can be found in the supplementary document -Section 7.

---

### Author Response · Authors · 2024-12-03

Dear Reviewers,

We would like to ask if our rebuttal has addressed all the questions.
We are glad to answer if there are more questions.

---

### Author Response · Authors · 2024-12-04

We sincerely thank all the reviewers for their insightful feedback and valuable suggestions, which have greatly contributed to improving the quality of this paper.

---

### Meta-Review · Area_Chair_n8by · 2024-12-20

**Metareview:**

In this paper, the authors proposed a long-term 3D point tracking method with cost volum fusion, which is performed at each level using a transformer architecture. The method outperforms the important baseline by projecting 2D tracking into 3D. The reviewers have a discussion after rebuttal, and there are still some limitations of the paper. More comprehensive comparisons with SpatialTracker should be included, and it is still unconvinced why SpatialTrack is considered 2.5D. Also, while the camera poses are assumed known for the proposed method, it does not substantially surpass methods without using camera poses. Due to the reasons, I recommend a decision of rejection.

**Additional Comments On Reviewer Discussion:**

Initially the reviewers raised concerns on the comparisons with SpatialTracker, the novelty and contributions (if it is really the first method to achieve 3D point tracking without test-time optimization), more experimental comparisons and evaluations, and discussions. The authors addressed some of them, but there are still important issues as I summarize in the metareview. Therefore I recommend a decision of rejection.

---

### Decision · Program_Chairs · 2025-01-22

Reject